# Estimating the Specific Yield and Groundwater Level of an Unconfined Aquifer Using Time-Lapse Electrical Resistivity Imaging in the Pingtung Plain, Taiwan

Ding-Jiun Lin [1], Ping-Yu Chang [1,*], Jordi Mahardika Puntu [1], Yonatan Garkebo Doyoro [1,2], Haiyina Hasbia Amania [1] and Liang-Cheng Chang [3]

1    Department of Earth Sciences, National Central University, Taoyuan 320, Taiwan
2    Department of Applied Geology, School of Natural Sciences, Adama Science and Technology University, Adama P.O. Box 1888, Ethiopia
3    Department of Civil Engineering, National Yang Ming Chiao Tung University, Hsinchu 300, Taiwan
*    Correspondence: pingyuc@ncu.edu.tw; Tel.: +886-3-422-7151 (ext. 65623)

**Abstract:** This study aims to apply geophysical methods to determine the Specific Yield (*Sy*) and Groundwater Level (GWL) in an unconfined aquifer of the Pingtung Plain in South Taiwan. *Sy* is an important hydraulic parameter for assessing groundwater potential. Obtaining specific yield for a large area is impractical due to the limited coverage and the high cost of the pumping test, which limits the potential evaluation of regional groundwater. Therefore, we used time-lapse Electrical Resistivity Imaging (ERI) to determine the *Sy* and GWL. Seasonal variations were considered when measuring time-lapse resistivity for five different months in 2019. We calculated the *Sy* and GWL from inverted resistivity data using empirical formulas and the soil–water characteristic curve (SWCC). We first used Archie's law to calculate the relative saturation change with depth for each ERI profile, and then we used the Van Genuchten (VG) and Brooks–Corey (BC) empirical equations to estimate *Sy* and GWL. Finally, we compared the obtained GWL to the existing observation well to verify the findings of our study. The results showed that the VG and BC are able to predict *Sy* and GWL; however, the BC result is less consistent with the observation well result. In the study area, the dry season GWL ranged from 24.5 m to 35.2 m for the VG results and from 25.7 m to 35.5 m for the BC results. The wet season GWL ranged from 26.5 m to 38.9 m for the VG and from 26.4 m to 38.2 m for the BC results. The spatial distribution of the GWL shows a high gradient of GWL in the northeastern region, induced by significant proximal fan recharge. The determined spatial distribution of *Sy* varies from 0.15 to 0.21 for the VG and 0.14 to 0.20 for the BC results, indicating the study area has significant potential for groundwater resources. Therefore, nondestructive resistivity imaging can be used to aid in the determination of hydraulic parameters.

**Keywords:** electrical resistivity imaging; groundwater; specific yield; Van Genuchten; Brooks–Corey; soil water characteristic curve

## 1. Introduction

The characterization of hydraulic properties at the aquifer scale is required for evaluating and managing groundwater resources. Understanding hydraulic parameters, particularly the specific yield of unconfined aquifers for a given region can help quantitatively evaluate the groundwater potential. Specific yield is well-studied theoretically; however, quantifying it using traditional pumping tests and applying it to determine a wide range of groundwater availability remains challenging [1,2]. One factor explaining this limitation is the complex nature of drainage and drawdown during pumping tests in heterogeneous unconfined aquifers, as well as delayed yield effects in the vadose zone [3]. Another difficulty in quantifying the specific yield from pumping tests is the scarcity and reliability of the available data, which results in limited specific yield calculations. A more comprehensive

pumping test with at least two wells (pumping and monitoring wells) and a longer pumping duration are commonly required for estimation. Noninvasive geophysical techniques, particularly electrical resistivity imaging (ERI), can help to estimate the hydraulic parameter of an unconfined aquifer more efficiently due to lower cost and higher convenience compared to constructing many monitoring wells [4–7].

Early work for hydrologically based electrical resistivity studies focused on locating potential groundwater sites. In recent years, it has also been used to estimate aquifer hydraulic parameters using empirical equations that associate resistivity and hydraulic parameters [5,7]. The well-known Archie equation has been widely used to determine the relationship between resistivity and porosity [4]. Hydraulic conductivity can be determined from resistivity measurements as the porosity and the hydraulic conductivity are related by empirical equations. By incorporating the Archie–Kozeny (AK) model, an empirical correlation between the resistivity and the hydraulic conductivity of an aquifer was also used [6]. Furthermore, the combined Archie's and Van Genuchten equation relates electrical resistivity and hydraulic conductivity to the degree of saturation [5].

The physical relationships between water content and suction in the vadose zone are described by the Van Genuchten (VG) [8] and Brooks–Corey (BC) empirical equations [9]. These empirical equations' performances may vary depending on the residual water content, saturated water content, and shape of the parameters. Despite the fact that several studies used empirical equations to determine hydraulic parameters, none of the studies evaluated which model is the best suited for calculating hydraulic parameters. Because the VG and BC appear promising for correlating unsaturated zones and unconfined aquifers, they are used to estimate the hydraulic parameters in this study.

Several studies have utilized geophysical methods to determine the hydraulic parameters of an unconfined aquifer. Tijani et al. [10] and de Almeida et al. [11] applied vertical resistivity sounding (VES) to estimate the hydraulic conductivity variation in porous aquifers. Vogelgesang et al. [12] used electrical resistivity tomography and hydraulic conductivity data to evaluate the relationship between electrical resistivity and hydraulic conductivity as input data for groundwater modeling in a local shallow alluvial aquifer. Boucher et al. [13] found an association between specific yield and transmissivity with pumping results and magnetic resonance sounding data. Using temporal gravity surveys, Blainey et al. [14] calculated specific yield and storage variations in managed groundwater recharge sites in northeastern Colorado. Tizro et al. [15] used the Frohlich and Parke [16] formula to determine the specific yield comparable to that obtained from pumping tests based on vertical electrical resistivity data. Farzamian et al. [17] and Kaleris and Ziogas [18] used time-lapse electrical resistivity imaging (ERI) to monitor saturation curve variation and estimate hydraulic conductivity using the VG or Kozeny–Carman empirical equation. Chang et al. [19] estimated hydraulic conductivities using time-lapse ERI during a pumping test. Even though they analyzed the hydraulic parameters, none of these studies examined how well the soil–water characteristic curve (SWCC) predicted the hydraulic parameters, mostly focusing on the specific location within one survey line or a small-scale study. Thus, the present study estimated the specific yield as well as groundwater level for a region in different seasons utilizing Electrical Resistivity Imaging and two different empirical equations (Brooks–Corey and Van Genuchten).

Most of the proximal fan in Taiwan consists of unconsolidated and semiconsolidated sand and gravel, largely under unconfined conditions. Understanding and characterizing aquifer properties can enable more effective resource management in Taiwan, where water demand is rapidly increasing. Due to the scarcity of borehole records and limited pumping tests, we used nondestructive resistivity imaging to estimate hydraulic properties in Pingtung Plain's proximal fan.

Groundwater Level (GWL) monitoring and hydraulic characteristics of the Pingtung proximal fan were determined using in-situ time-lapse ERI. The data were collected using the Wenner and Schlumberger array due to its high signal-to-noise ratio and sensitivity to lateral and vertical variations [20]. We conducted an ERI survey over a five-month

period to evaluate the impact of seasonal rainfall recharges on GWL variation. We used Archie's Law to calculate the Van Genuchten (VG) and Brooks–Corey (BC) parameters after incorporating time-lapse ERI data into SWCC. The obtained VG and BC parameters are the residual water content, saturated water content, suction, and shape parameters ($\alpha$, $n$, and $\lambda$), which are used to estimate the Specific Yield ($Sy$) and GWL. We have calculated the $Sy$ and GWL of the Pingtung proximal fan to assess the groundwater resources. The research provides insights into how to use empirical equations to determine hydraulic parameters.

## 2. Study Area

In Taiwan, the Pingtung alluvial plain is one of the potential groundwater aquifers formed by the Gaoping, Donggang, and Linbian rivers. According to Central Geology Survey (CGS) drilling and geophysical exploration results [21], the depth of the bedrock basement is shallow in the northern Pingtung Plain and deeper in the southern part. The bedrock overlies the Pleistocene Lingkou conglomerate and Quaternary alluvial deposits (Figure 1). The primary formation of the study area aquifers is composed of Holocene alluvial deposits' gravel and sand layers with varying sediment thicknesses ranging from a few meters to over a hundred meters, which are highly permeable to groundwater movement. The study area is situated between the proximal fan and partly in the middle fan. The proximal fan of sandy gravel sediment primarily acts as the recharging zone and is characterized as an aquifer unit. According to the Sipu–Taishan (SP–TS) hydrogeological profiles, the upper part of the middle fan, which mostly consists of sandy gravel, contributes to groundwater recharge and is also classified as an unconfined aquifer unit, whereas the lower part of the middle fan with silty and clayey sand is classified as a confining unit (Figure 2). For further hydrogeological studies, we refer the reader to the Central Geology Survey, Taiwan [22].

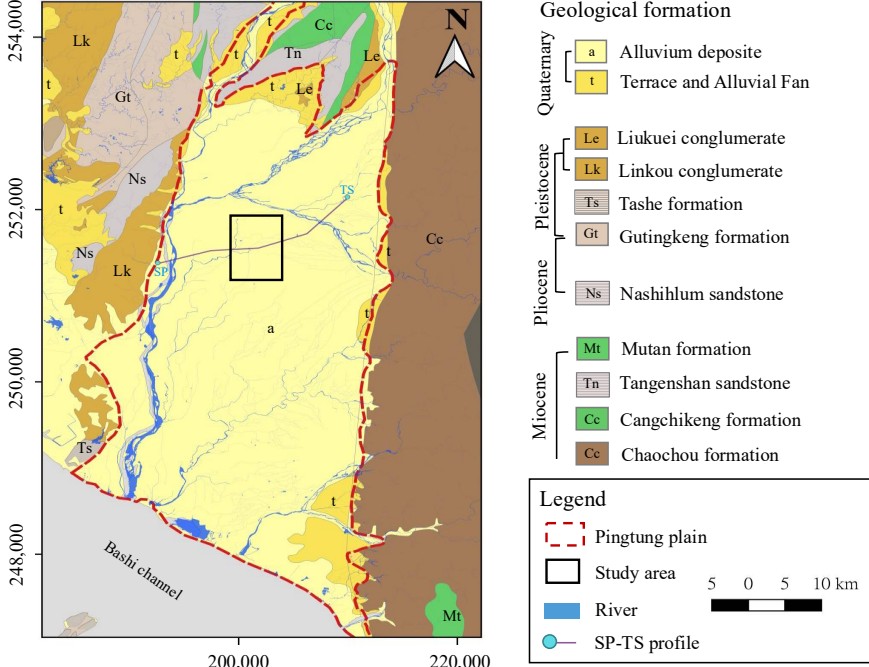

**Figure 1.** The geological map of the Pingtung plain after the Central Geological Survey [23]. The rectangular square represents the study area and the circle with a line is SP–TS hydrogeological profile.

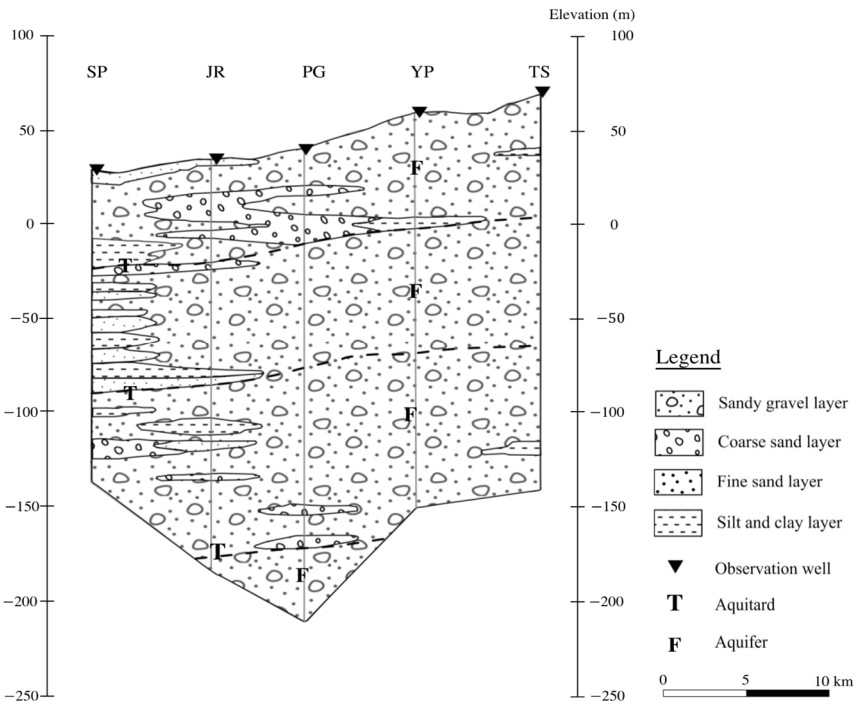

**Figure 2.** SP–TS hydrological profile of the northern Pingtung Plain. The five observation wells in the profile are represented as SP: Sipu, JR: Jiuru, PG: Pengcuo, YP: Yanpu, and TS: Taishan. The dashed line is considered the boundary of the aquifers.

According to the Taiwan Central Weather Bureau data, the study area received 20.5 mm of precipitation in February, 78.5 mm in April, 806.5 mm in July, 191.0 mm in September, and 3.0 mm in November. Figure 3 depicts the daily precipitation records for the Yanpu meteorological station, which is the nearest station to the study area. The rainy season in the Pingtung Plain typically lasts from mid-May until the end of September. Figure 3 also shows the daily groundwater level (GWL) of the Pengcuo (PG) observation well, which is located in the center of the study area.

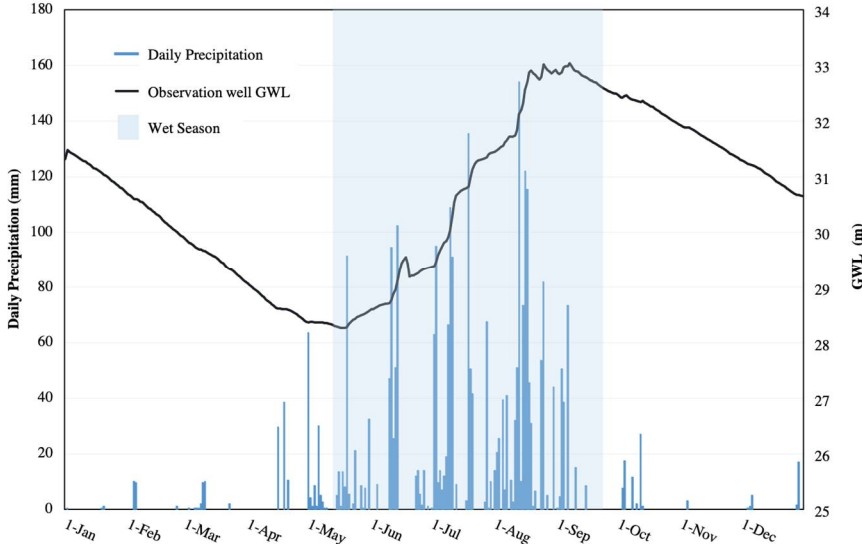

**Figure 3.** Taiwan Central Weather Bureau meteorological observatory daily precipitation record and GWL of PG observation well.

## 3. Materials and Methods

### 3.1. Electrical Resistivity Imaging Data Acquisition and Processing

We conducted a time-lapse Electrical Resistivity Imaging (ERI) survey at 10 sites (Figure 4) in February, April, July, September, and November 2019. The resistivity data were measured using a Lippmann 4-point light 10 W resistivity meter [24]. We used the Wenner and Schlumberger configuration, with 101 electrodes spaced 1.5 m apart. We deployed 1 survey station, PT01, nearby the Pengcuo (PG) observation well to verify the reliability of resistivity data in monitoring GWL. The ERI survey sites PT02, PT03, PT05, PT06, PT07, and PT08 are located to the north of the PG observation well, while PT04, PT09, and PT10 are located to the south. Because of local agricultural activity and nearby artificial noises, the resistivity data of PT01 and PT10 for February, PT09 for May, PT08 for July, and PT05 for September were not included in this study analysis.

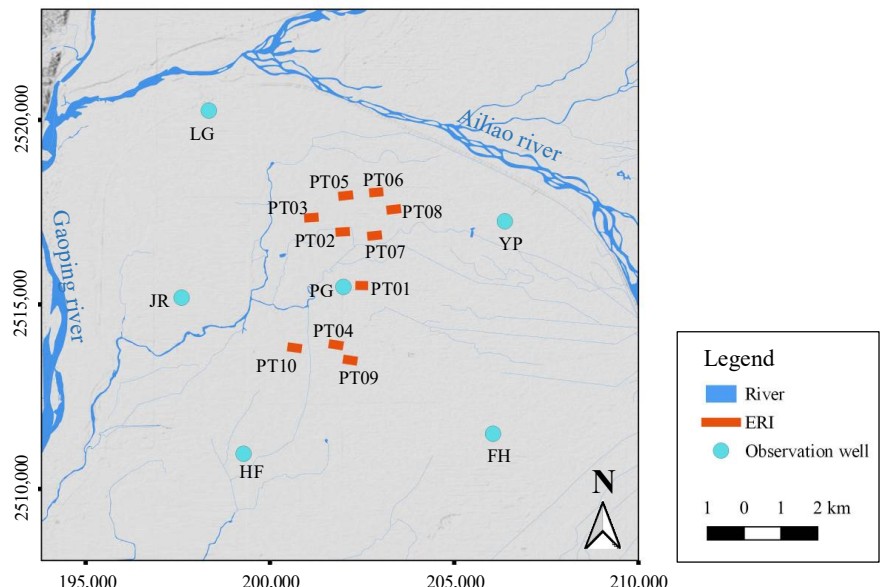

**Figure 4.** ERI survey location and groundwater observation wells. HF: Haifeng; JR: Jiuru; LG: Ligang; PG: Pengcuo; YP: Yanpu; FH: Fanhua.

Furthermore, we processed and analyzed the resistivity data obtained from the field survey with several steps as shown in Figure 5. First, we inverted the resistivity data with the software EarthImager2D^TM (Version 2.4.2) [25]; in this step, we applied a smoothness-constrained least-square inversion method as it helps to reduce the squares of the spatial variations in model resistivity, which is more appropriate for settings with gradually changing resistivity to obtain the 2-D geoelectric model [26]. Second, we extracted at least 5 sets of 1-D vertical inverted resistivity data from each 2-D inversion result. Then, we converted the 1-D inverted resistivity data to normalized water content by applying Archie's law. Finally, we estimated the groundwater level (GWL) and theoretical specific yield (*Sy*) using Van Gecnuchen (VG) and Brooks–Corey (BC) empirical equations. The spatial distribution of GWL and *Sy* was interpolated using the ordinary kriging method. Additionally, we used GWL data from other observation wells near the study region as boundary constraints. The spatial distribution data were then imported into QGIS software to create a contour map.

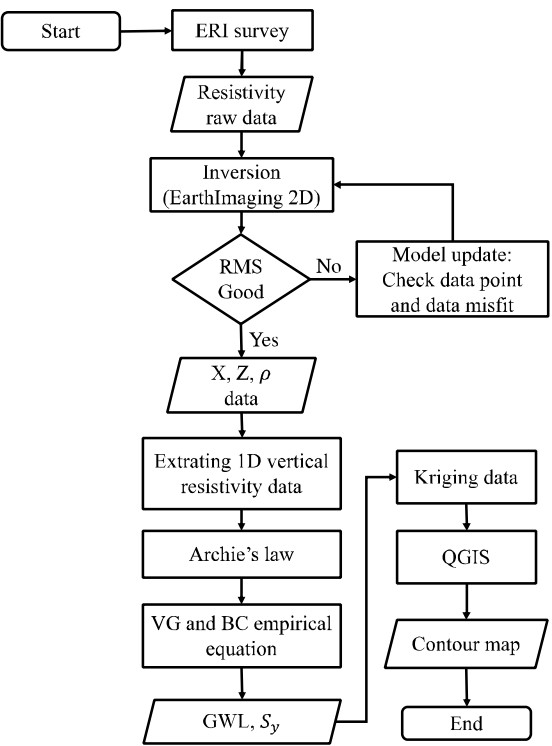

**Figure 5.** Workflow of the study.

### 3.2. Empirical Estimation of Hydraulic Parameter from Resistivity Data

We utilized Archie's law [4] to empirically estimate the water content from at least 5 1-D vertical resistivity data in the central part of each resistivity survey line. Several physical factors may affect the resistivity measurements, including water content, porosity, pore structure, and cementation of material. Archie's law relates resistivity and sediment saturation using the following expression:

$$\rho = \alpha\rho_w\phi^{-n}S_w^{-m} \tag{1}$$

where $\rho$ is the resistivity of the material, $\rho_w$ is the pore water resistivity, $\alpha$ is the tortuosity factor, $\phi$ is the porosity, $S_w$ is the saturation, and $m$ and $n$ are the saturation index and cementation index, respectively.

Applying the approach used by Dietrich et al. [27] and Chang et al. [28] that links the unsaturated and saturated resistivities and shows resistivity variation with the water saturation, we determined normalized volumetric water content, $\Theta$:

$$\Theta = \phi_A S_r \tag{2}$$

where $S_r$ is normalized relative saturation and $\phi_A$ is the average porosity of the soil. Che, et al. [29] analyzed soil grain size in Pingtung Plain and determined the porosity using the Vokovic and Soro [30] empirical equation. The result showed that the Pingtung Plain porosity ranges from 0.26 to 0.30, whereas the porosity of the Pengcuo area varies from 0.26 to 0.27. Thus, this study used the average porosity $\phi_A$ of 0.26.

According to the drilling records, the lithology of the uppermost 1 to 2 m consists of surface soil, characterized by clay and sand. Therefore, we eliminated the uppermost 2 m as they represent surface soil layer features rather than gravel aquifer properties. Figure 6 shows an example of the normalized volumetric water content variation curve with depth extracted from the 1-D vertical resistivity set of PT06 during the April survey, and the variation curve has a similar trend to the Soil Water Characteristic Curve (SWCC). Utilizing the SWCC, we can estimate the GWL and specific yield.

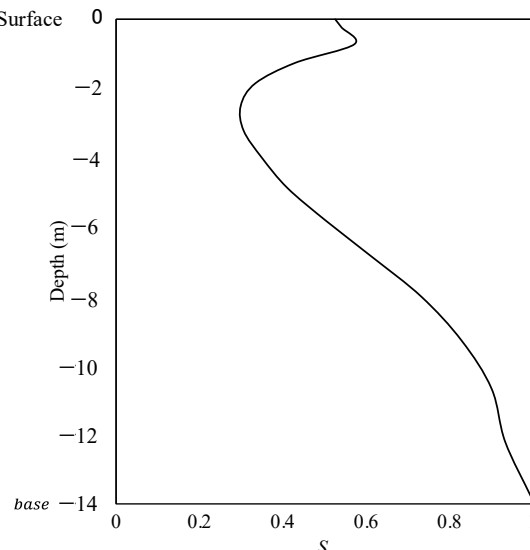

**Figure 6.** The normalized relative saturation ($S_r$) variation with depth from the PT06 during the April survey.

Several studies have applied the SWCC [31,32], and the most frequently used of empirical equations are the Brooks–Corey (BC) and the Van Genuchten (VG). The BC can describe the relationships between the suction and water contents in the vadose zone:

$$\theta(h) = \begin{cases} \theta_r + (\theta_s - \theta_r)\left(\frac{h_a}{h}\right)^{\lambda}, h_a < h \\ \theta_s, h_a \geq h \end{cases} \tag{3}$$

where $\theta(h)$ represents unsaturated water content ($L^3L^{-3}$), $\theta_r$ represents the residual water content ($L^3L^{-3}$), $\theta_s$ indicates the saturated water content ($L^3L^{-3}$) with the presumed average porosity $\phi_A$, $h_a$ represents the air-entry suction head (L), $h$ represents the suction head (L), and $\lambda$ indicates the pore size distribution parameter.

The relationship between water content and the suction head was also obtained using the VG:

$$\Theta(h) = \theta_r + \frac{\theta_s - \theta_r}{\left[1 + (\alpha h)^n\right]^m} \tag{4}$$

where $\Theta(h)$ represents the normalized water content ($L^3L^{-3}$), $\alpha$ is related to the inverse of air entry value, $n$ is related to the pore size distribution of the soil, and $m$ is associated with the asymmetry of the model [33], where $m$ equals $1 - n^{-1}$.

The Brooks–Corey (BC) and Van Genuchten (VG) provide a functional relationship to describe the unsaturated soil property in the vadose zone and are widely applied in the SWCC [31,34]. Figure 7 depicts the VG with a continuous SWCC. The VG has 3 shape parameters ($\alpha$, m, and $n$); these factors enable more flexibility, allowing the equation to better fit the data for different soil types. The BC has a sharp boundary that corresponds to a capillary head. Because BC has 2 parameters (h and a), obtaining them from BC is simpler than obtaining the VG parameters. To determine hydraulic parameters in the SWCC, this study used both the VG and BC. We inverted the BC and VG parameters by minimizing the root mean square differences between estimated and measured water content. The resistivity measurement yields the measured water content. To optimize the minimum object equation in the inversion, we utilized the Generalized Reduced Gradient (GRG) nonlinear program [35] with the EXCEL Solver. The Solver is an optimization tool in EXCEL that can be used to determine how changing the assumptions in a model can lead to the desired outcome. This tool can be accessed from the add-ins menu in EXCEL [36,37].

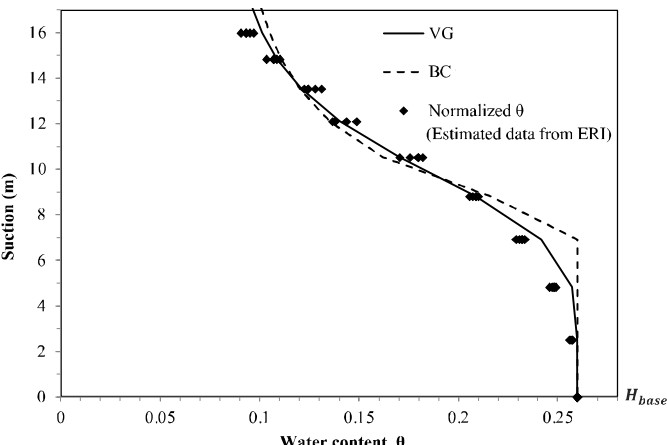

**Figure 7.** Soil–Water Characteristic Curve (SWCC) for 5 selected vertical profiles of the normalized volumetric water content from 1-D vertical resistivity of PT06 is indicated by a diamond symbol. The dashed curve represents the fitted BC results to the normalized volumetric water content obtained from resistivity measurements, while the solid curve indicates the fitted VG results.

The Groundwater Level (GWL) in the study area was estimated using an approach based on the assumption that the saturated layer is a base with a depth of $H_{base}$. Additionally, we assumed that the suction head is proportional to the height of the saturated zone in the unconfined aquifer, as discussed in Krahn and Fredlund [38]. Then, we estimated the air-entry suction head where water content is in a saturated condition ($\theta = \theta_s$) relative to the same base, $h_{top}$. As a result, we were also able to estimate the depth of the saturated surface of the air-entry suction head, $D$.

$$D = H_{base} - h_{top}(\theta = \theta_s) \qquad (5)$$

The actual groundwater table, however, may differ from the surface that corresponds to the air-entry suction. Therefore, we applied the correction factor, $K_s$, to correct the predicted groundwater depth to the groundwater depth obtained from an observation well. Then GWL was obtained by subtracting the groundwater depth ($D$) from the meter above sea level (MASL). The following is an illustration of the correction equation.

$$D = H_{base} - h_{top}(\theta = K_s\theta_s) \qquad (6)$$

Furthermore, we quantified the theoretical specific yield, $S_y$, which represents the average volume of water that can be drained per unit aquifer surface per unit drop of the head. It is an essential hydraulic parameter to describe the groundwater potential of a particular area. Therefore, the $Sy$ was calculated using the difference between the SWCC saturated water content $\theta_s$ and residual water content $\theta_r$:

$$S_y = \theta_s - \theta_r \qquad (7)$$

## 4. Results

### 4.1. Time-Lapse ERI Survey

The time-lapse Electrical Resistivity Imaging (ERI) studies were carried out at 10 sites in the Pingtung Plain over five months (February, April, July, September, and November) as described in Section 3. Figures 8 and 9 illustrate the inverted models we obtained for four stations (PT02, PT07, PT04, and PT06) to show the resistivity of seasonal change. During the dry seasons, there was a large fluctuation in resistivity from the surface to the bottom of the model section. For example, the inverted model from April indicated 20 to 50 Ohm-m for topsoil with a 2 m thickness. The second layer, with resistivities ranging from 100 to 2000 Ohm-m and depths ranging from 2 to 14 m is interpreted as unsaturated sand and

gravel. The bottom layer's resistivity gradually decreases over a depth of 14 m, starting from 100 to 20 ohm-m. Topsoil with a 1.5 m thickness demonstrates 10 to 30 Ohm-m during the wet season. A layer of unsaturated sandy gravel with a resistivity of 80 to 300 Ohm-m can be found in July with a layer thickness of 5 m. However, because of the significant rainfall, it is no longer visible in the September geoelectric profile. During the wet season, the bottom layer beneath 7 m in depth had a resistivity of less than 80 Ohm-m. The lowest layer's resistivity is much lower than in the dry season, indicating that the water table is significantly higher in the rainy season.

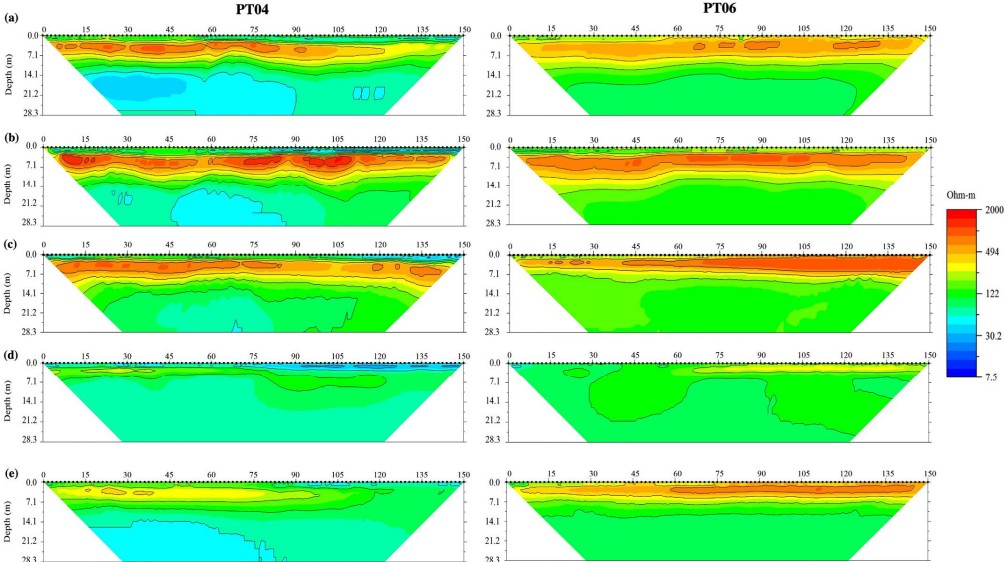

**Figure 8.** Time-lapse inverted resistivity models of PT04 and PT06 sites for (**a**) February, (**b**) April, (**c**) July, (**d**) September, and (**e**) November.

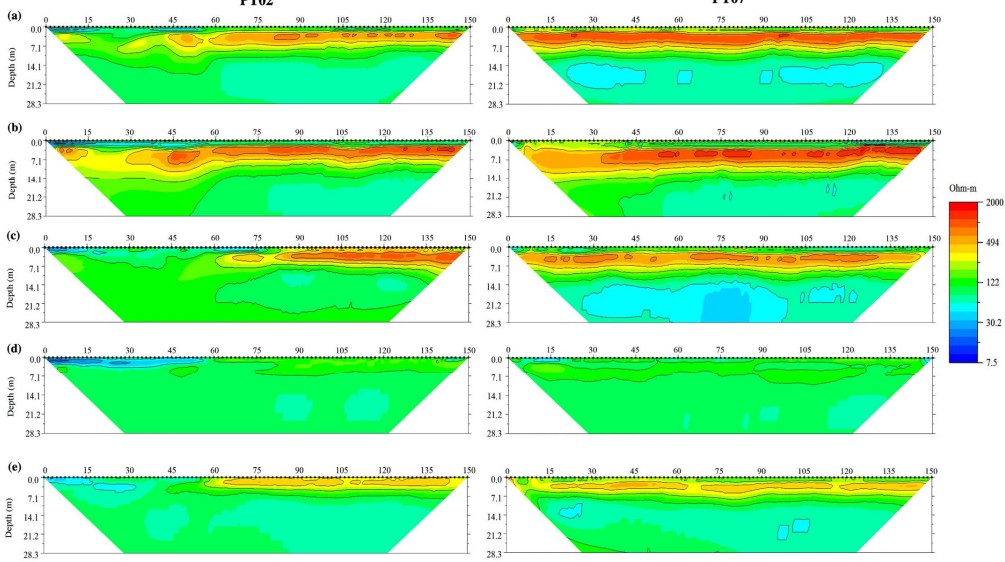

**Figure 9.** Time-lapse inverted resistivity models of PT02 and PT07 sites for (**a**) February, (**b**) April, (**c**) July, (**d**) September, and (**e**) November.

### 4.2. Soil–Water Characteristic Curve (SWCC) and Groundwater Level (GWL)

To determine the relative hydraulic parameters and GWL, we used Van Genuchten (VG) and Brooks–Corey (BC) of SWCC for an unconfined aquifer. The normalized water content, which is determined from ERI data, was plotted as the function of the suction head in SWCC. It is well-fitted with both VG and BC. Tables 1 and 2 show the fitting parameters for the VG and BC, respectively, for the PT06 site as an example. In the wet season, the VG

parameter *n*, which is connected to the pore size distribution index, has a higher value than in the dry season. The BC pore size distribution parameters, so-called λ, have also shown a similar trend with the *n* parameter of VG for dry and wet seasons.

**Table 1.** The estimated relative parameter of Brooks–Corey from the time-lapse ERI survey at the PT06 site for five different months of 2019.

| VG Parameters | February | April | July | September | November |
|---|---|---|---|---|---|
| $\alpha$ | 8.46 | 11.04 | 10.69 | 10.73 | 9.73 |
| $n$ | 4.29 | 4.01 | 9.19 | 6.49 | 7.65 |
| m | 0.77 | 0.75 | 0.89 | 0.85 | 0.87 |
| $\theta_r$ | 0.08 | 0.05 | 0.09 | 0.08 | 0.08 |
| $\theta_s$ | 0.26 | 0.26 | 0.26 | 0.26 | 0.26 |

**Table 2.** The estimated relative parameter of Brooks–Corey from the time-lapse ERI survey at the PT06 site for five different months of 2019.

| BC Parameters | February | April | July | September | November |
|---|---|---|---|---|---|
| $\lambda$ | 1.81 | 1.35 | 3.68 | 1.23 | 3.03 |
| $h_a$ | 5.85 | 6.79 | 8.64 | 8.63 | 7.48 |
| $\theta_r$ | 0.08 | 0.07 | 0.13 | 0.18 | 0.08 |
| $\theta_s$ | 0.26 | 0.26 | 0.26 | 0.26 | 0.26 |

Figures 10 and 11 demonstrate the SWCC of VG and BC for representative sites of PT2, PT04, PT06, and PT07 throughout the survey period. Generally, The VG and BC showed similar trends in SWCC. The inverse of the air-entry value, which is relative to VG parameter α, influences the VG. The VG curve moves up for smaller α, indicating the wet season (Figures 10 and 11). On the other hand, the VG curve shifts downward for greater α, indicating the dry season. Both the VG and BC showed a high air-entry height during the wet season compared to the dry season. The VG and BC curves of PT02 and PT07, however, did not follow the proper SWCC, which could be due to highly moist site conditions or a perched aquifer; this issue will be addressed in future studies. Because of the abrupt change in the BC, it has a larger air-entry value than the VG, although the VG has a continuous SWCC. The VG may, therefore, be more realistic for determining GWL under in-situ conditions.

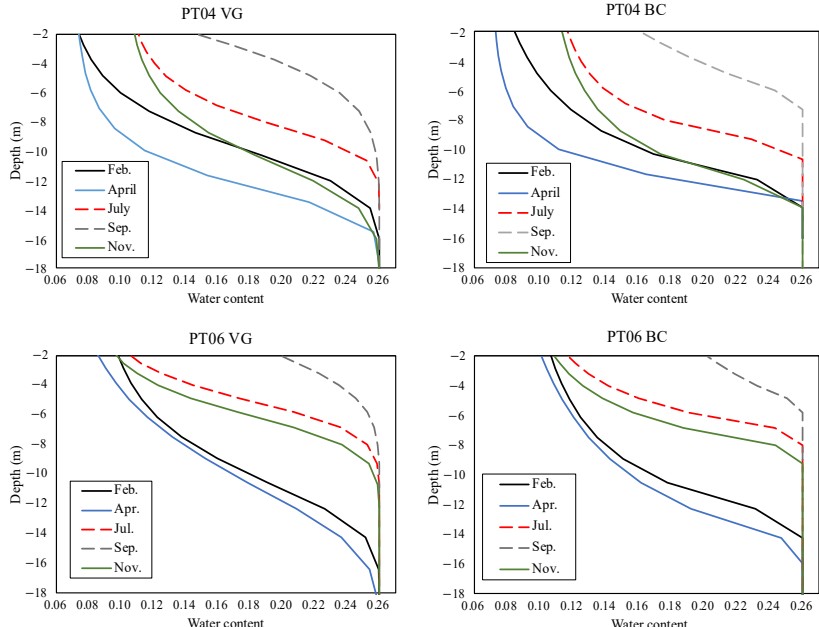

**Figure 10.** The Van Genuchten (VG) and Brooks–Corey (BC) for PT04 and PT06 sites for five different months. The dashed curve corresponds wet season while the solid curve represents the dry season.

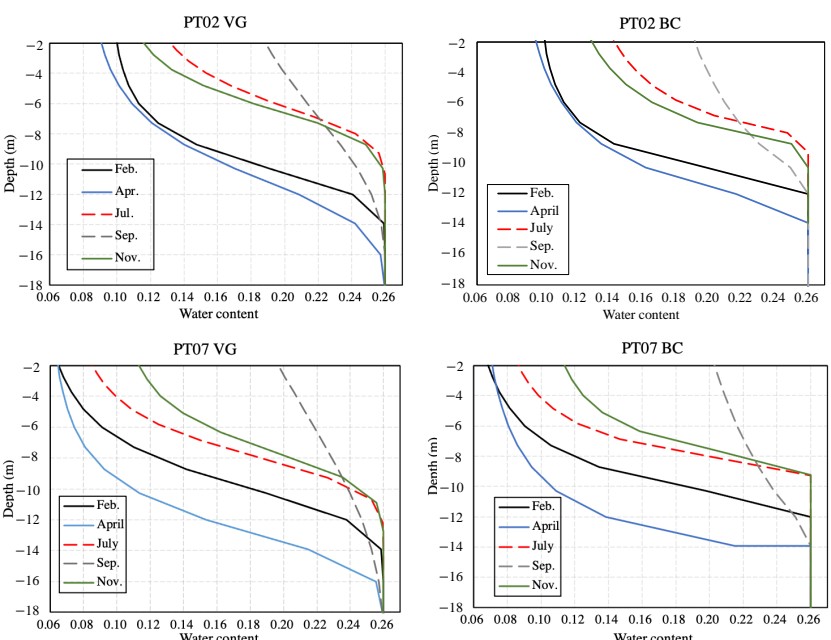

**Figure 11.** The Van Genchten (VG) and Brooks–Corey (BC) for PT02 and PT07 sites for five different months. The dashed curve corresponds to the wet season while the solid curve represents the dry season.

We determined the GWL using SWCC. We applied a correction factor based on the GWL record of PG observation well. We applied the correction factor ($K_s = 0.91$) in Equation (6) to obtain a more reliable GWL. The estimated GWL showed the best fitting with observation well results after a correction factor was applied, as shown in Tables 3 and 4. The estimated GWL of the study area is shown to be generally smallest in April, and it gradually increases in the wet season. For instance, the VG and BC for site PT06 indicated 29 m and 28.4 m in April and then increased to 38.9 m and 37.7 m in September. After the wet season (November), the GWL decreased to 35 m in the VG and 35.9 m in the BC.

**Table 3.** The estimated Groundwater Level derived from Van Genchten (VG) for five different months of 2019. (MASL: Meters Above Sea Level).

| Sites/Well | MASL | Groundwater Level (m) | | | | |
| --- | --- | --- | --- | --- | --- | --- |
| | | February | April | July | September | November |
| PT01 | 41.9 | - | 27.1 | 30.0 | 33.4 | 32.5 |
| PT02 | 40.0 | 28.2 | 26.5 | 32.3 | 30.8 | 32.0 |
| PT03 | 37.0 | 28.4 | 26.5 | 33.2 | 29.2 | 31.8 |
| PT04 | 38.7 | 26.3 | 24.5 | 29.0 | 29.9 | 25.7 |
| PT05 | 41.0 | 29.8 | 27.9 | 36.0 | - | 35.2 |
| PT06 | 43.0 | 30.0 | 29.0 | 36.1 | 38.9 | 35.0 |
| PT07 | 40.0 | 28.1 | 25.3 | 30.3 | 30.5 | 30.7 |
| PT08 | 45.1 | 31.4 | 30.6 | - | 37.5 | 34.0 |
| PT09 | 40.4 | 27.4 | - | 28.7 | 28.8 | 25.2 |
| PT10 | 34.0 | - | 24.5 | 27.9 | 26.5 | 22.2 |
| Well | 40.2 | 30.8 | 28.1 | 30.5 | 33.1 | 32.1 |

After obtaining the Groundwater Level (GWL) of each ERI site, we utilized the ordinary kriging with the point kriging type. This method assumes that the mean of the variable is constant across the study area, and the point kriging estimates the values of the points at the grid nodes to interpolate the spatial distribution of the GWL. As a boundary constraint, we also applied GWL data from other observation wells near the study region. The spatial distribution data were imported into QGIS software to generate a contour map. Figures 12 and 13 illustrate the spatial distribution of the GWL generated from VG and BC, respectively. Both revealed comparable trends in groundwater distribution. The groundwater in the studied area usually flows from northeast to southwest. However, the groundwater gradient in the

studied area varies significantly between the wet and dry seasons. The dry season (February and April) exhibited a substantially larger gradient in the northeastern region of the research area. During the rainy season (July and September), groundwater gradients were higher in the northeastern than in the southwestern side, especially in September. The steep gradient in the northeast direction is most likely due to significant recharging in the proximal fan, which is characterized by coarse-grained sediments. In contrast, high-gradient groundwater was observed on the southwestern side in November, which could be related to the increased groundwater flow from the proximal fan to the middle fan. Section 5.1 discusses the GWL variance for the dry and wet seasons in depth.

**Table 4.** The estimated Groundwater Level derived from Brooks–Corey (BC) for five different months of 2019.

| Sites/Well | MASL | Groundwater Level (m) | | | | |
|---|---|---|---|---|---|---|
| | | February | April | July | September | November |
| PT01 | 41.9 | - | 28.7 | 31.5 | 30.3 | 33.2 |
| PT02 | 40.0 | 29.9 | 28.3 | 32.6 | 30.0 | 31.9 |
| PT03 | 37.0 | 29.6 | 28.3 | 33.2 | 30.4 | 32.4 |
| PT04 | 38.7 | 27.3 | 27.1 | 29.9 | 26.4 | 27.1 |
| PT05 | 41.0 | 29.9 | 29.5 | 36.9 | - | 34.1 |
| PT06 | 43.0 | 31.3 | 29.7 | 36.5 | 38.2 | 35.5 |
| PT07 | 40.0 | 29.9 | 26.8 | 32.0 | 28.2 | 32.3 |
| PT08 | 45.1 | 33.4 | 32.0 | - | 38.1 | 35.0 |
| PT09 | 40.4 | 28.9 | - | 28.7 | 27.0 | 27.0 |
| PT10 | 34.0 | - | 25.7 | 27.4 | 23.9 | 21.2 |
| Well | 40.2 | 30.8 | 28.1 | 30.5 | 33.1 | 32.1 |

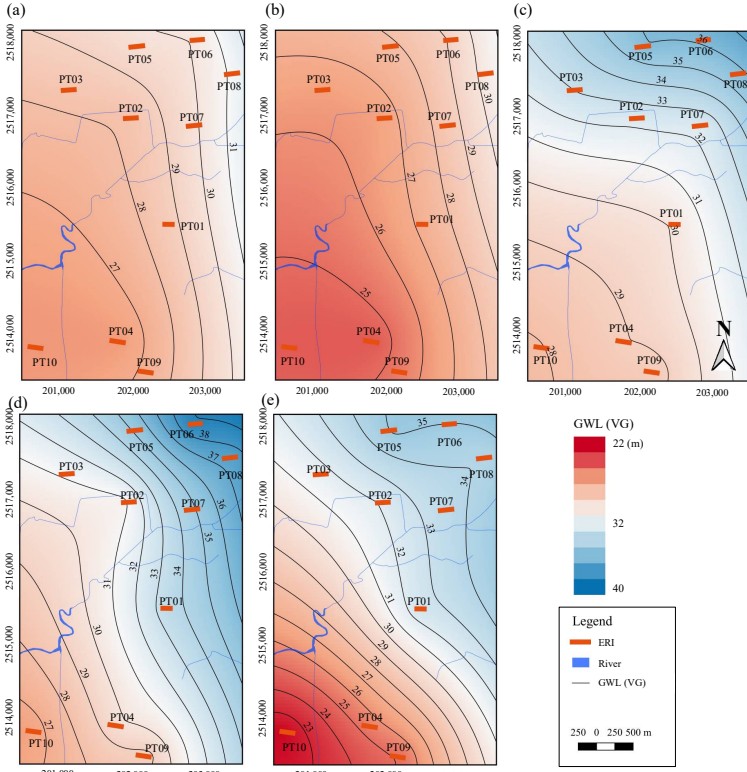

**Figure 12.** Spatial distribution contour map of the GWL obtained from VG for different months: (**a**) February, (**b**) April, (**c**) July, (**d**) September, and (**e**) November.

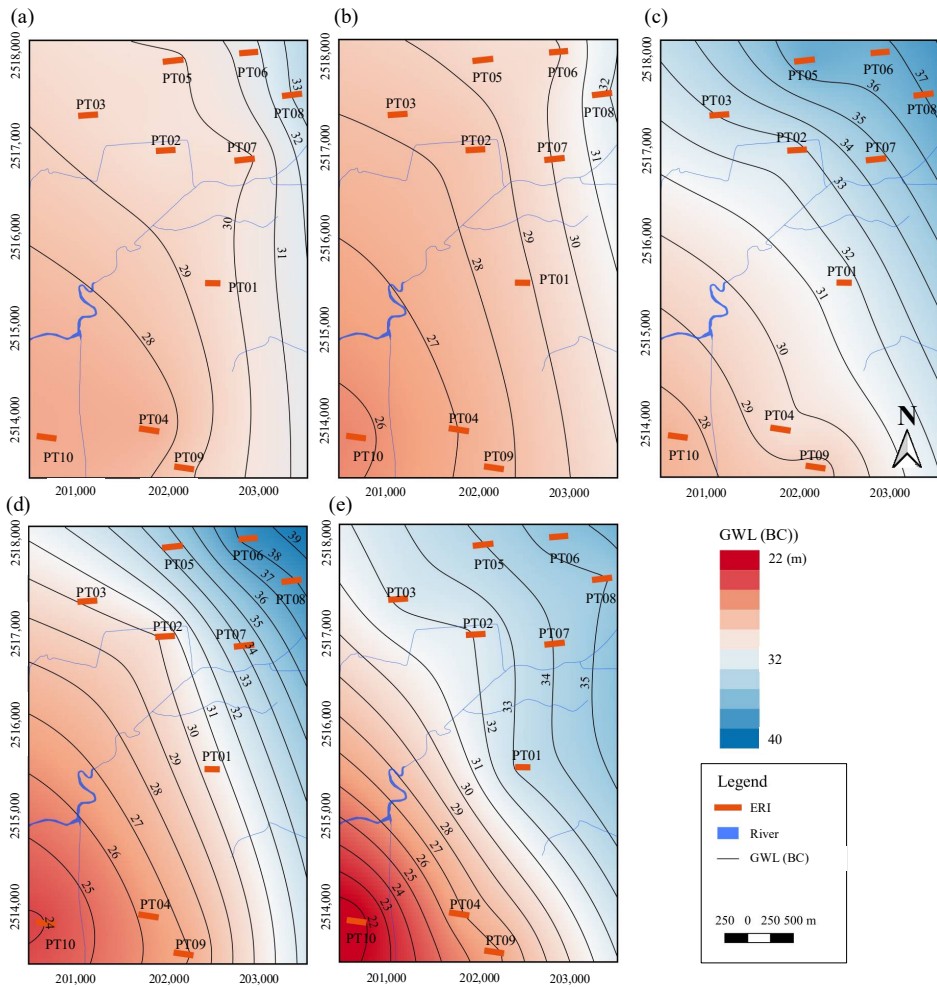

**Figure 13.** Spatial distribution contour map of the GWL obtained from BC for different months:
(**a**) February, (**b**) April, (**c**) July, (**d**) September, and (**e**) November.

### 4.3. Theoretical Specific Yield

We also calculated the theoretical specific yield ($Sy$) using Equation (7) for the unconfined aquifer of Pingtung Plain. Tables 5 and 6 showed theoretical specific yield ($Sy$) derived from the Van Genchten (VG) and Brooks–Corey (BC) of the survey data. For the wet season, the $Sy$ obtained from the VG ranged from 0.07 to 0.21, whereas it ranged from 0.04 to 0.19 in BC. In contrast, the $Sy$ of the dry season ranged from 0.15 to 0.21 for the VG and from 0.14 to 0.20 for the BC, demonstrating that, compared to the wet season, the dry season has higher $Sy$. Figure 14 presented the $Sy$ spatial distribution for April; it indicated that $Sy$ was the highest in the northeastern part of the research area and decreased to the southwestern part of the study area for both the VG and BC. To verify the results, we used the $Sy$ obtained from the pumping test result of the PG well, which is close to ERI site PT01. The calculated $Sy$ for the dry season at the PT01 location is 0.18 for the VG and 0.19 for the BC. According to the Central Geology Survey [22], the Pengcuo well-pumping test yielded 0.173 specific yield, which is more consistent with the VG result.

**Table 5.** The estimated theoretical specific yields (*Sy*) were calculated from Van Genuchten for five different months of 2019.

| Site | Theoretical Specific Yields (*Sy*) | | | | | Max. *Sy* |
|---|---|---|---|---|---|---|
| | February | April | July | September | November | |
| PT01 | - | 0.18 | 0.18 | 0.07 | 0.18 | 0.18 |
| PT02 | 0.17 | 0.20 | 0.15 | 0.12 | 0.16 | 0.20 |
| PT03 | 0.21 | 0.20 | 0.16 | 0.14 | 0.21 | 0.21 |
| PT04 | 0.20 | 0.19 | 0.16 | 0.14 | 0.16 | 0.20 |
| PT05 | 0.18 | 0.21 | 0.21 | - | 0.18 | 0.21 |
| PT06 | 0.18 | 0.21 | 0.18 | 0.13 | 0.18 | 0.21 |
| PT07 | 0.21 | 0.19 | 0.19 | 0.18 | 0.16 | 0.21 |
| PT08 | 0.21 | 0.21 | - | 0.16 | 0.21 | 0.21 |
| PT09 | 0.19 | - | 0.20 | 0.12 | 0.18 | 0.20 |
| PT10 | - | 0.15 | 0.14 | 0.14 | 0.13 | 0.15 |

**Table 6.** The estimated theoretical specific yields (*Sy*) were calculated from Brooks–Corey for five different months of 2019.

| Site | Theoretical Specific Yields (*Sy*) | | | | | Max. *Sy* |
|---|---|---|---|---|---|---|
| | February | April | July | September | November | |
| PT01 | - | 0.19 | 0.19 | 0.07 | 0.18 | 0.18 |
| PT02 | 0.16 | 0.17 | 0.13 | 0.07 | 0.14 | 0.17 |
| PT03 | 0.16 | 0.19 | 0.14 | 0.04 | 0.16 | 0.19 |
| PT04 | 0.18 | 0.19 | 0.15 | 0.13 | 0.24 | 0.19 |
| PT05 | 0.15 | 0.19 | 0.19 | - | 0.08 | 0.19 |
| PT06 | 0.16 | 0.19 | 0.16 | 0.08 | 0.16 | 0.19 |
| PT07 | 0.20 | 0.19 | 0.18 | 0.06 | 0.17 | 0.20 |
| PT08 | 0.20 | 0.18 | - | 0.09 | 0.12 | 0.20 |
| PT09 | 0.17 | - | 0.18 | 0.09 | 0.17 | 0.18 |
| PT10 | - | 0.14 | 0.04 | 0.05 | 0.02 | 0.14 |

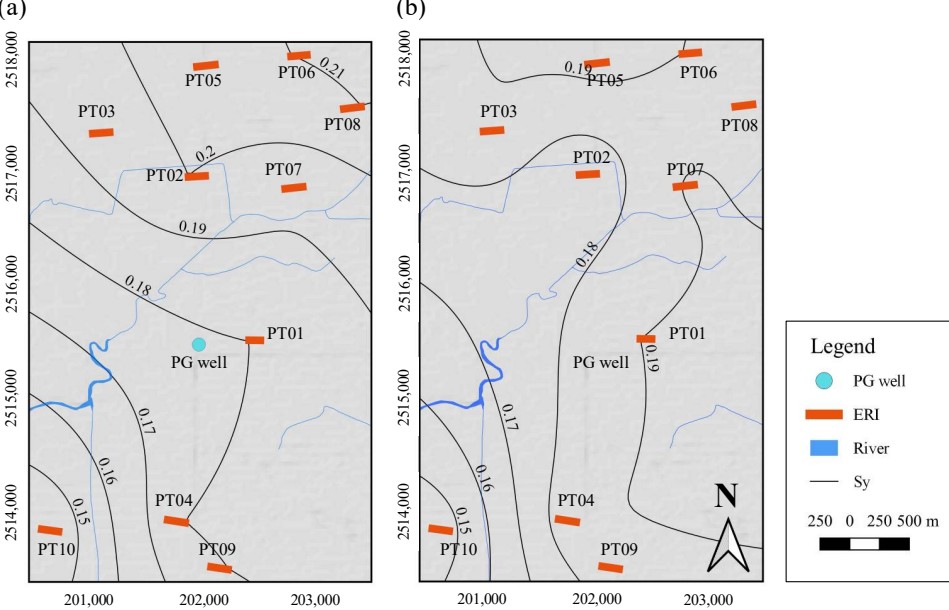

**Figure 14.** Spatial distribution of theoretical specific yield for April using (**a**) Van Genuchten, and (**b**) Brooks–Corey.

## 5. Discussion

### 5.1. Groundwater Level Difference for Wet and Dry Seasons

We plotted the spatial distribution of the Groundwater Level (GWL) difference for the wettest (September) and the driest (April) months using the Van Genuchten (VG), as shown in Figure 15. The GWL of the study area significantly rises in September due to high

precipitation. Specifically, the northeastern part of the study area has a significant GWL difference due to the groundwater inflow from the Central Mountain Ridge and Ailio River water recharging, in line with other studies [32,39]. The lower GWL difference is shown in the southern part of the study area. We found that the local artifactual activity may also influence the GWL variation, such as PT04, which is near the fishing pond. During the dry season, groundwater nearby the PT04 site is frequently discharged to fill the fishing pond, which decreases the water level and results in a relatively larger GWL difference than the surrounding neighboring sites.

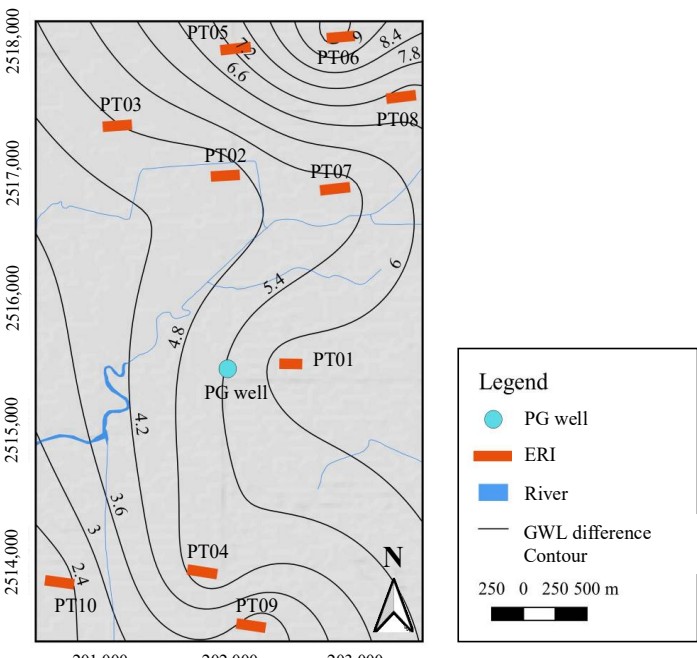

**Figure 15.** The spatial distribution of the Van Genuchtan groundwater level difference for the wet (September) and dry (April) seasons.

### 5.2. Groundwater Distribution and Paleochannel

In general, the groundwater distribution of the study flows from northeast to southwest, which corresponds to the groundwater flow model of the Pingtung Plain built by Chang et al. [40] and Ting et al. [41]. The groundwater model is based on the current observation wells records and the current river distribution. However, the distribution of groundwater in the Pingtung Plain may also be influenced by other local structures, such as paleochannels [42,43]. In the study area, the river system had been changed due to flooding, and after several riverbank stabilizations, formed the current river system. Figure 16 shows the overlay of the wet season groundwater level onto the 1904 paleochannel map. The GWL of the VG results indicates groundwater flows from northeast to southwest, which corresponds to the direction of the Wuluo paleochannel flow. This suggests that local structures, such as paleochannels, make a significant contribution to the groundwater flow direction of the unconfined aquifer in the Pingtung Plain.

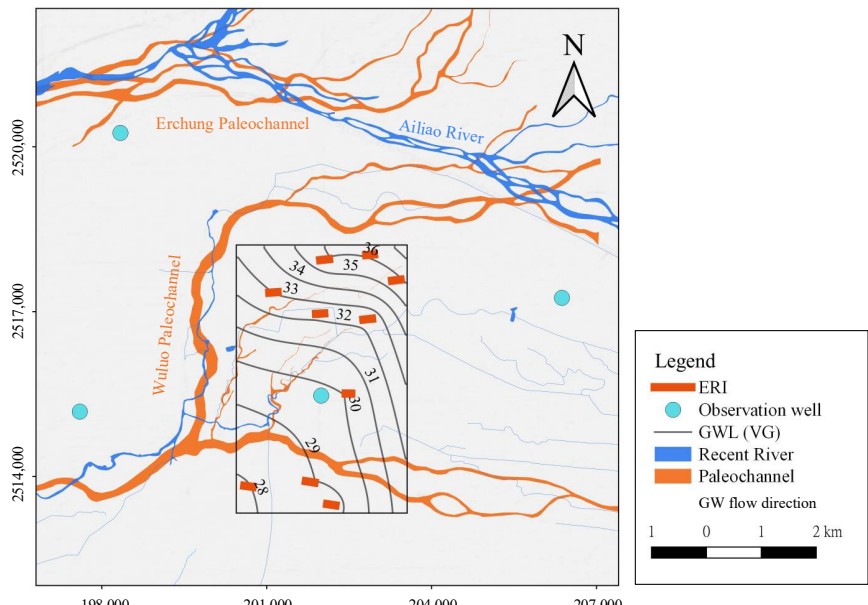

**Figure 16.** The distribution of the Groundwater Level (GWL) in wet season overlays the paleochannel map of the 1904 year.

### 5.3. SWCC Efficiency in Determining Hydraulic Parameters

The different Soil–Water Characteristic Curve (SWCC) empirical equations have been used to calculate hydraulic parameters. The hydraulic parameters are effectively determined using the Van Genchten (VG) and Brooks–Corey (BC), which link inverted resistivity to porosity. We calculated the theoretical specific yield (*Sy*) and groundwater level (GWL) for the unconfined aquifer in Pingtung Plain.

In general, both the VG and BC produce consistent groundwater distribution. However, the GWL generated from the BC is higher than that derived from the VG (Figure 17), particularly during the dry season. The high GWL of the BC in the dry season is due to its sensitivity to the capillary fringe zones, which are thicker in the dry season than in the wet season. Despite the correction factor, the BC curve showed a high suction head and high GWL. In contrast, the VG is less sensitive to the capillary fringe and its suction head after the correction factor adequately determines the GWL, which is well correlated to the nearby observation well GWL for both dry and wet seasons. Therefore, the VG and BC can sufficiently determine the GWL using nondestructive time-lapse resistivity imaging data, yet the VG should be prioritized over the BC since the VG is considered to be a more accurate and reliable equation for describing SWCC and predicting soil hydraulic properties than the BC.

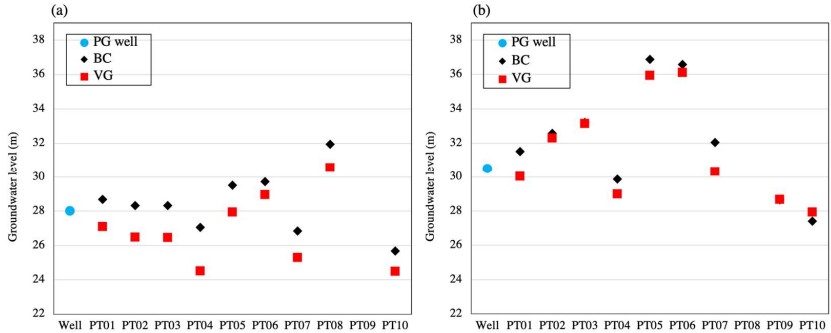

**Figure 17.** Estimated Groundwater Level comparisons of the Van Genchtan (VG) and Brooks–Corey (BC) for (**a**) dry season and (**b**) wet season.

Moreover, the VG and BC have properly determined the specific yield ($Sy$) of the study area. Generally, high $Sy$ is obtained in the northeastern part of the study area, which is related to the coarse-grained material of the proximal fan as the coarse sediment has a high groundwater storage capacity. In contrast, the low $Sy$ in the southwestern area is associated with the medium-grained sediments of the middle fan. In addition, the $Sy$ varied for dry and wet seasons, showing lower $Sy$ in the wet season than in the dry season. This is likely due to the variation between the wet and dry season's hysteresis of the drying and wetting curve of the SWCC, consistent with Chang et al. [28]. The $Sy$ is efficiently calculated by both the VG and BC. The VG result is more reliable and consistent with the specific yield obtained from the pumping test of the nearby observation well. One limitation of this study could be addressed in future research. The GWL correction factor is adjusted based on the one represented observation well, and the assumption that the hydrogeological formation is similar, as mentioned in Section 2. This may not fully consider the region's heterogeneity, so conducting further observations would reduce this problem. Despite the limitations of this study, our findings are still capable of monitoring the variation of the GWL in different seasons and the hydrogeological characteristic parameters for the large study area.

## 6. Conclusions

We used an alternative approach using time-lapse Electrical Resistivity Imaging (ERI) to examine the hydrogeological characteristics and groundwater distribution of the unconfined aquifer in the Pingtung Plain. We conducted 10 ERI in different locations over five different months in 2019. Based on the Van Genuchten (VG) and Brooks–Corey (BC) empirical equations, we estimated the Groundwater Level (GWL) and theoretical Specific Yield ($Sy$) as the preliminary evaluation index of the groundwater resource.

The ERI time-lapse results revealed three distinct geoelectric layers. As topsoil, the initial layer resistivity ranged from 20 to 50 Ohm-m. The resistivity of the second layer ranged from 100 to 2000 Ohm-m, indicating unsaturated sandy gravel. The resistivity of the bottom layer decreases with depth from 100 to 20 Ohm-m, which is associated with saturated sandy gravel.

The GWL of the research area is effectively determined by the soil–water characteristic curve (SWCC) of the BC and VG. The distribution of groundwater is consistent in both the VG and BC. In the dry season, the GWL ranged from 24.5 m to 35.2 m in the VG results, whereas the results of BC varied from 25.7 m to 35.5 m. The wet season GWL for the VG ranged from 26.5 m to 38.9 m, whereas the BC ranged from 26.4 m to 38.2 m. The groundwater in the study area flows from northeast to southwest. In addition to the Pingtung Plain's precipitation, the groundwater distribution is governed by river recharge and groundwater flow from the Central Mountain Ridge. The paleochannel in the study area may also influence the distribution of groundwater flow. The high gradient of GWL in the northeastern region is induced by significant proximal fan recharge.

Furthermore, the $Sy$ of the study area is adequately calculated by the BC and VG. The $Sy$ ranges determined by the VG are 0.15 to 0.21, while the $Sy$ ranges obtained by the BC are 0.14 to 0.20. The maximum $Sy$ values are taken for survey sites. The coarse-grained sediments of the proximal fan contribute to the high $Sy$ in the studied area. According to the obtained $Sy$, the Pingtung Plain has good potential for groundwater exploitation.

Both the VG and BC are sufficient for determining hydraulic parameters. However, the VG is thought to be a more accurate equation than the BC for describing SWCC and predicting soil hydraulic properties. Our study results show that the VG delivered more reliability in determining groundwater level and $Sy$ compared to the BC. Therefore, the VG should be prioritized when analyzing time-lapse ERI resistivity data. This study provides insight into how to apply a nondestructive geophysical method to quantify groundwater potential in an unconfined aquifer.

**Author Contributions:** Conceptualization, D.-J.L., P.-Y.C. and J.M.P.; Data curation, D.-J.L., J.M.P. and H.H.A.; Formal analysis, D.-J.L.; Funding acquisition, P.-Y.C. and L.-C.C.; Investigation, D.-J.L., J.M.P., Y.G.D. and H.H.A.; Methodology, P.-Y.C., J.M.P. and D.-J.L.; Project administration, D.-J.L., and

P.-Y.C.; Resources, D.-J.L., P.-Y.C. and L.-C.C.; Supervision, P.-Y.C. and L.-C.C.; Visualization, D.-J.L. and Y.G.D.; Writing—original draft, D.-J.L.; Writing—review & editing, D.-J.L., P.-Y.C., Y.G.D., J.M.P. and H.H.A. All authors have read and agreed to the published version of the manuscript.

**Funding:** This research was funded by the Central Geological Survey of the Ministry of Economy, R.O.C. (Taiwan), under the project "The Investigation of Hydrogeology and Groundwater Resources—The Utilization Improvement and Capacity Assessment of Underground Reservoir (3/4)" (Project Number: 108-5226904000-01-02-01).

**Data Availability Statement:** The data are available upon request to the authors.

**Conflicts of Interest:** The authors declare that they have no competing interest.

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
