# Peer review of "Estimating the Specific Yield and Groundwater Level of an Unconfined Aquifer Using Time-Lapse Electrical Resistivity Imaging in the Pingtung Plain, Taiwan"

_water, doi:10.3390/w15061184_

Round 1

Reviewer 1 Report

See attached file

Reviewer 2 Report

The manuscript “Estimating the specific yield and groundwater level of an unconfined aquifer using time-lapse electrical resistivity imaging in the Pingtung Plain, Taiwan” presents a non-destructive technique to determine hydraulic parameters. This study is valuable in reducing the cost of the traditional hydraulic test. However, appropriate revisions to the following points should be undertaken in order to justify the recommendation for publication.

1. The author of “Liang-Cheng Chang” does not appear in the list of authors.

2. Line 48-51, please provide appropriate references for this statement.

3. In the section Introduction, the authors need to address the issue about if the proposed research approach to determine the specific yield and groundwater level is new.

4. Line 171-175, Chen et al. [26] evaluated the relationship between K and the grain size of the area. How did this study determine porosity? Pengtsuo area in the Pingtung Plain? The author should rewrite the sentences to make it clear and logical.

5. Line 184, several “researches” should change to several “studies”.

6.  In figure 6, please address how to obtain the estimated data in Figure 6 more explicitly.

7.  Concerning the spatial distribution contour map of the GWL, how did the contour map generate?

8.  Finally, grammatical errors, punctuation errors, and spelling errors should be carefully examined throughout the entire manuscript.

Reviewer 3 Report

1. Introduction section is weak, add more recent papers 

2. Study area details should be match with the objective of the present study Refer.

3. I didnt find the methodology section in the present study

4. Add the flow chart to clear about the methodology 

5. Discussion section is too weak, add more description about thee results

6. Elaborate the results findings in the section 5

7. Add more reference related to this study, 

8. Revise the conclusion based on the correction carried out in the main text

Round 2

Reviewer 1 Report

The manuscript has been improved substantially. The authors did a good work to address a multitude of issues made by the referees. However the final version still requires some clarifications, as some parts of the workflow are still not sufficiently outlined. 

At the end of page 7 (line 401ff) they talk about the inversion of the SWCC curve, which relates suction to water content. The suction is derived from the  resistivity, as it is stated briefly in the text (using equation (1)?). But what about the suction? The conjugate gradient method is a method to solve linear systems. How are they used in the EXCEL solver for an optimization problem? There is not something like the EXCEL solver! This passage requires further extensions and clarifications! 

Similarly: line 584. Thank you for mentioning kriging as method to step from point data to areal representations. However, there are various kriging methodologies and programs. Can you go more into detail, in order to be able to evaluate your methodology?    

Reviewer 3 Report

Accept

Author Response

Dear reviewer,

We sincerely thank the reviewer for the suggestions and comments in our manuscript. We highly appreciated the in-depth review and constructive suggestions. Have a great day.

Best regards,

Authors